# What is my quantum computer good for? Quantum capability learning with physics-aware neural networks

**Daniel Hothem**
Quantum Performance Laboratory
Sandia National Laboratories
Livermore, CA 94550
dhothem@sandia.gov

**Ashe Miller**
Quantum Performance Laboratory
Sandia National Laboratories
Albuquerque, NM 87185
anmille@sandia.gov

**Timothy Proctor**
Quantum Performance Laboratory
Sandia National Laboratories
Livermore, CA 94550
tjproct@sandia.gov

## Abstract

Quantum computers have the potential to revolutionize diverse fields, including quantum chemistry, materials science, and machine learning. However, contemporary quantum computers experience errors that often cause quantum programs run on them to fail. Until quantum computers can reliably execute large quantum programs, stakeholders will need fast and reliable methods for assessing a quantum computer's capability—i.e., the programs it can run and how well it can run them. Previously, off-the-shelf neural network architectures have been used to model quantum computers' capabilities, but with limited success, because these networks fail to learn the complex quantum physics that determines real quantum computers' errors. We address this shortcoming with a new quantum-physics-aware neural network architecture for learning capability models. Our scalable architecture combines aspects of graph neural networks with efficient approximations to the physics of errors in quantum programs. This approach achieves up to $\sim 50\%$ reductions in mean absolute error on both experimental and simulated data, over state-of-the-art models based on convolutional neural networks, and scales to devices with 100+ qubits.

## 1 Introduction

Quantum computers have the potential to efficiently solve classically intractable problems in quantum chemistry [Cao et al., 2019], materials science [Rubin et al., 2024], machine learning [Harrow et al., 2009], and cryptography [Shor, 1997]. While contemporary quantum computers are approaching the size and noise levels needed to solve interesting problems [Arute et al., 2019], they are far from being capable of reliably running most useful quantum programs [Proctor et al., 2021a]. Until we build quantum computers capable of executing *any and all* useful and interesting quantum programs, stakeholders will require fast, reliable, and scalable methods for predicting the programs that a given quantum computer can reliably execute.

The task of learning which quantum programs a particular quantum computer can reliably execute is known as *quantum capability learning* [Proctor et al., 2021a]. Quantum capability learning is very

38th Conference on Neural Information Processing Systems (NeurIPS 2024).

difficult because the number of possible (Markovian) errors plaguing a quantum computer grows exponentially in its size [Blume-Kohout et al., 2022], i.e., in the number of qubits ($n$) it contains, and errors in a quantum program can combine in difficult-to-predict ways [Proctor et al., 2021a]. Most existing approaches to capability learning restrict themselves to learning how well a quantum computer executes a small set of quantum programs, by running all of those programs and estimating a success metric for each one [Lubinski et al., 2023, Proctor et al., 2024]. While these methods provide insight into a quantum computer's capability, they are not predictive.

Recently, several groups have proposed building predictive models of a quantum computer's capability using convolutional neural networks (CNNs) [Elsayed Amer et al., 2022, Hothem et al., 2024b, Vadali et al., 2024, Hothem et al., 2023b] and graph neural networks (GNNs) [Wang et al., 2022]. However, these neural-network-based capability models achieve only modest prediction accuracy when applied to real quantum computers, because they fail to learn the complex physics that determines real quantum computers' failures [Hothem et al., 2024b].

In this work, we introduce a novel quantum-physics-aware neural network (qpa-NN) architecture for quantum capability learning (Fig. 1). Our approach uses neural networks with GNN-inspired structures to predict the rates of the most physically relevant errors in quantum programs. These predicted error rates are then combined using an efficient approximation to the exact (but exponentially costly) quantum physics formula for how those errors combine to impact a program's success rate. Our approach leverages the graph structures that encode the physics of how errors' rates typically depend on both the quantum program being run and how a quantum computer's qubits are arranged, and it offloads the difficult-to-learn, yet classically tractable task of approximately combining these error rates to predict a circuit's performance to an already-known function. This enables our qpa-NNs to vastly outperform the state-of-the-art CNNs of Hothem et al. [2024b] on both experimental and simulated data without sacrificing the ability to model large devices of 100+ qubits.

Our qpa-NNs are enabled, in part, by focusing on learning a quantum computer's capability on high-fidelity quantum programs, which are those programs that a quantum computer correctly executes with high probability. High-fidelity programs are arguably the most interesting programs to study as we care far more about whether a quantum computer successfully executes a program 99% or 90% of the time rather than 1% or 10% of the time.

In a head-to-head comparison, our qpa-NNs achieve a $\sim 50\%$ reduction in mean absolute error (MAE) over the CNNs of Hothem et al. [2024b], on average and on the same experimental datasets. Our qpa-NNs achieve an average $\sim 36\%$ improvement over those CNNs even after fine-tuning those CNNs on the same subset of the training data (high-fidelity programs) used to train our qpa-NNs.

Our qpa-NNs' improved performance is likely largely due to their improved ability to model the impact of coherent errors on a program's success rate. Off-the-shelf networks struggle with coherent errors [Hothem et al., 2024b], but our qpa-NNs are designed to model how these errors add up and cancel out, making the qpa-NNs much better predictors in the presence of coherent errors. To verify this, we demonstrate that our qpa-NNs can accurately predict the performance of random circuits run on a hypothetical 4-qubit quantum computer experiencing only coherent errors. Our qpa-NN obtained a $\sim 50\%$ lower MAE than a CNN, averaged across five datasets, and the trained qpa-NN even exhibits moderate performance when making predictions for a different class of circuits (random mirror circuits [Proctor et al., 2021a]) simulated on the same hypothetical 4-qubit quantum computer, i.e., our qpa-NNs display moderate prediction accuracy on out-of-distribution data.

We make the following contributions in our work:

1. We introduce qpa-NNs, a bespoke neural network architecture for modeling the capability of a quantum computer, which outperform state-of-the-art CNN models by $\sim 50\%$ on experimental and simulated data.

2. We use our qpa-NNs to model the capability of a simulated 100-qubit device; the largest ever neural network capability learning demonstration by a factor of two.

3. We demonstrate, for the first time, how to train NNs to predict the process fidelity [Nielsen, 2002] of a circuit, which is the most widely used quantum channel error metric.

4. We provide evidence that the improved performance of our qpa-NNs is partly due to their ability to better model the effect of coherent errors, which are known to be challenging for other state-of-the-art methods.

## 2 Background

In this section, we review the background in quantum computing necessary to understand this paper. See Nielsen and Chuang [2010] for an in-depth introduction to quantum computing and Blume-Kohout et al. [2022] or Hashim et al. [2024] for a thorough description of the errors in quantum computers.

### 2.1 Quantum computing

A quantum computer performs computations using qubits, which are two-level systems whose pure states are unit vectors in a complex two-dimensional Hilbert space, $\mathcal{H}$. The pure states of $n$ qubits are unit vectors in $\mathcal{H}^{\otimes n}$. The two orthonormal vectors $|0\rangle$ and $|1\rangle$ that are eigenvectors of the $Z$ Pauli operator are identified as the *computational basis* of $\mathcal{H}$. Errors and noise in real quantum computers mean that they are typically in states $\rho$ that are probabilistic mixtures of pure states.

A quantum computation is performed by running a quantum program, typically known as a *quantum circuit* (see illustration in Fig. 1a). An $n$-qubit quantum circuit ($c$) of depth $d$ is defined by a sequence of $d$ layers of logical instructions $\{L_i\}$. Executing $c$ consists of preparing each qubit in $|0\rangle$, applying each $L_i$, and then measuring each qubit to obtain an $n$-bit string $b$. Each layer typically consists of parallel one- and two-qubit gates, and it is intended to implement a $2^n \times 2^n$ unitary $U(L_i)$. Together, the layers are intended to implement $U(c) = U(L_d) \cdots U(L_1)$.

If quantum circuit $c$ is implemented without error, its output bit string $b$ is a sample from a distribution $P(c)$ whose probabilities are given by $\Pr(b = x) = |\langle x|U(c)|00\cdots0\rangle|^2$ where $|00\cdots0\rangle = |0\rangle \otimes \cdots \otimes |0\rangle$, $|x\rangle = |x_1\rangle \otimes \cdots \otimes |x_n\rangle$, and $x_i$ is the $i$-th bit in $x$. However, when a circuit is executed on a real quantum computer, errors can occur and this means that its output bit string $b$ is a sample from some other distribution $\tilde{P}(c)$. The process of errors corrupting a quantum computation can be modelled as follows. Each logic layer $L_i$ implements the intended unitary superoperator $\mathcal{U}(L_i) :$ $\rho \to U(L_i)\rho U^\dagger(L_i)$, where $\rho$ is a general $n$-qubit state, followed by an error channel $\Lambda_i$ that is a completely positive and trace preserving (CPTP) superoperator [Blume-Kohout et al., 2022]. The imperfect implementation of a circuit $c$ is then simply $\tilde{\mathcal{U}}(c) = \prod_{i=1}^{d} \Lambda_i \circ \mathcal{U}(L_i)$, and the output bit string $b$ is $x$ with probability $\Pr(b = x) = \text{Tr}(|x\rangle\langle x|\tilde{\mathcal{U}}(c)[|00\cdots0\rangle\langle00\cdots0|])$.

### 2.2 Quantum capability learning

Because quantum computers are error-prone, knowing which quantum circuits a particular quantum computer can execute with low error probability is important. Known as *quantum capability learning* [Proctor et al., 2021a, Hothem et al., 2024b], this task formally involves learning the mapping between a set of quantum circuits $c \in \mathcal{C}$ and some success metric $s(c) \in \mathbb{R}$ quantifying how well $c$ runs on a quantum computer $\mathcal{Q}$. In this work, we consider a large class of circuits known as Clifford (or stabilizer) circuits [Aaronson and Gottesman, 2004], which are sufficient to enable quantum error correction [Campbell et al., 2017], and two widely-used success metrics: *probability of successful trial* (PST) (a.k.a. success probability) and the *process fidelity* (a.k.a. entanglement fidelity) [Hothem et al., 2024b, Nielsen, 2002].

PST is defined only for definite-outcome circuits, which are circuits whose output distribution has (if run without error) support on a single bit string, $b(c)$. For any such circuit $c$, PST is defined as

$$\text{PST}(c) = \Pr(\text{measuring } b(c) \text{ when executing } c \text{ on } \mathcal{Q}). \tag{1}$$

In practice, $\text{PST}(c)$ is estimated by running the circuit $N_{\text{shots}} \gg 1$ times on $\mathcal{Q}$ and calculating

$$\widehat{\text{PST}(c)} = \frac{\# \text{ observations of } b(c)}{N_{\text{shots}}}. \tag{2}$$

Process fidelity is defined for all circuits, and it quantifies how close the actual quantum evolution of the qubits is to the ideal unitary evolution. It is given by

$$F(c) = \frac{1}{4^n}\text{Tr}\left[\tilde{\mathcal{U}}(c)\mathcal{U}^{-1}(c)\right]. \tag{3}$$

Estimating $F(c)$ is more complicated than estimating $\text{PST}(c)$, but efficient methods exist, such as mirror circuit fidelity estimation [Proctor et al., 2022]. Hence, in theory, it is possible to efficiently gather training data using either $\text{PST}(c)$ or $F(c)$ on arbitrarily large quantum computers.

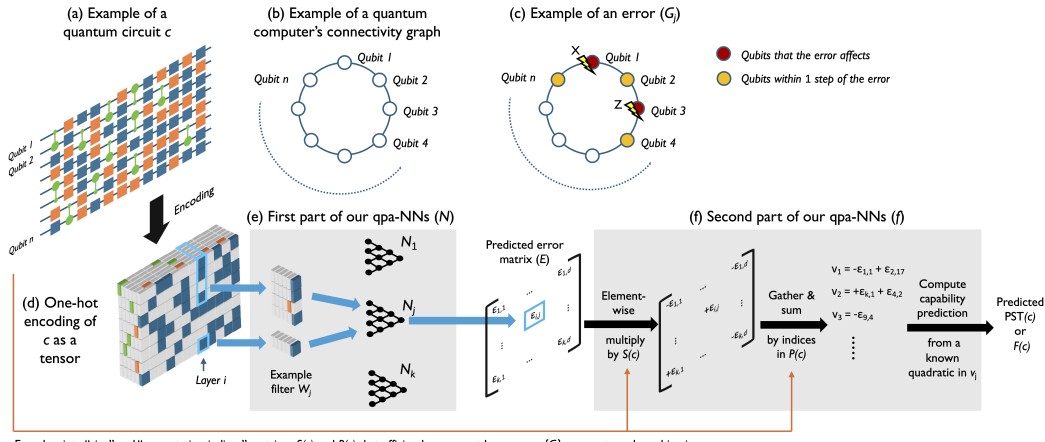

Figure 1: **Quantum capability learning with quantum-physics-aware neural networks (qpa-NNs)**. Our qpa-NNs are a novel architecture for learning a quantum computer's capability, i.e., the mapping from quantum circuits (or programs) to how well that imperfect quantum computer can run those circuits. These networks build in physical principles for how errors in quantum circuits occur—which can be expressed in terms of a quantum computer's connectivity graph—and efficient approximations to the physics of how these errors combine to impact a circuit's success rate.

## 2.3 Modelling errors in quantum computers

Our qpa-NNs build in efficient approximations to the quantum physics of errors in quantum computers. They do so using the following parameterization of an error channel: $\Lambda = \exp(\sum_j \epsilon_j G_j)$. Here $\mathbb{G}_n = \{G_j\}$ is the set of $2^{2n+1} - 2$ different Hamiltonian (H) and Stochastic (S) *elementary error generators* introduced by Blume-Kohout et al. [2022], and $\epsilon_j$ is the *rate* of error $G_j$. Not every kind of error process can be represented in this form (e.g., amplitude damping, or non-Markovian errors), but this parameterization includes many of the most important kinds of errors in contemporary quantum computers. Each H and S error generator is indexed by a non-identity element of the $n$-qubit Pauli group ($\mathbb{P}_n$). The Pauli operator indexing an H or S error indicates the qubits it impacts and its direction, e.g., the H error generator indexed by $X \otimes I^{\otimes(n-1)}$ is a coherent error on the 1st qubit and it rotates that qubit around its $X$ axis.

Our qpa-NNs use approximate formulas for computing $\text{PST}(c)$ or $F(c)$ from the rates of H and S errors, which we now review. Consider pushing each error channel $\Lambda_i$ to the end of the circuit and combining them together, i.e., we compute the error channel $\Lambda(c)$ defined by $\tilde{\mathcal{U}}(c) = \Lambda(c) \circ \mathcal{U}(c)$. Then

$$\text{PST}(c) \approx 1 - \sum_{P \in \mathbb{P}_n^{X,Y}} \left( s_P + \theta_p^2 \right), \tag{4}$$

where $s_P$ and $\theta_P$ are the rates of the $P$-indexed $S$ and $H$ error generators, respectively, in $c$'s error channel $\Lambda(c)$, and $\mathbb{P}_n^{X,Y}$ is the set of $n$-qubit Pauli operators containing at least one $X$ or $Y$. Similarly,

$$F(c) \approx 1 - \sum_{P \in \mathbb{P}_n} \left( s_P + \theta_p^2 \right). \tag{5}$$

Equations (4) and (5) are good approximations for low-error circuits [Mądzik et al., 2022]. However, they both suffer from the same flaw: they require tracking $\mathcal{O}(4^n)$ parameters. To address this problem, our qpa-NNs make an approximation: they only account for the contributions of a polynomially-sized set of errors that contains all those errors which are most likely to be experienced by a quantum computer. In this work, we chose to account for only local, low-weight errors, i.e., those with initial support on a small, connected subset of a device's connectivity graph.

# 3 Neural network architecture

Our neural network architecture (see Fig. 1) for quantum capability learning combines neural network layers that have GNN-like structures with efficient approximations to the physics of errors in quantum computers. The overall action of our neural networks is to map an encoding of a circuit $c$ to a prediction for $\mathrm{PST}(c)$ or $F(c)$. The same network can predict either $\mathrm{PST}(c)$ or $F(c)$ by simply toggling between two different output layers that have no trainable parameters. Our architecture is divided into two sequential parts. The first part of our architecture is a neural network $\mathcal{N}$ that has the task of learning about the kinds and rates of errors that occur in quantum circuits. We use GNN-like structures within $\mathcal{N}$ to embed physics knowledge for how those errors depend on the quantum circuit being run. The second part of our architecture is a function $f$ with no learnable parameters, that turns $\mathcal{N}$'s output into a prediction for $\mathrm{PST}(c)$ or $F(c)$.

## 3.1 Physics-aware neural networks for predicting errors in quantum circuits

The neural network $\mathcal{N}$'s input is a quantum circuit $c$ of depth $d(c)$ represented by (i) a tensor $I(c) \in \{0,1\}^{n \times d(c) \times n_{ch}}$ describing the gates in $c$ (see Fig. 1a), and (ii) a matrix $M(c) \in \{0,1\}^{2 \times n}$ describing the measurement of the qubits at the end of $c$. $\mathcal{N}$ maps $I(c)$ to a matrix $\mathcal{E} \in \mathbb{R}^{k \times d(c)}$ and $M(c)$ to a vector $\vec{m} \in \mathbb{R}^k$. $\mathcal{E}_{ij}$ is a prediction for the rate with which error type $j$ occurs during circuit layer $i$, and $m_j$ is a prediction for the rate with which error type $j$ occurs when measuring the qubits at the end of a circuit. There are $2(4^n - 1)$ different possible error types that can occur in principle (see Section 2) so it is infeasible to predict all their rates beyond very small $n$. However, the overwhelming majority of these errors are implausible, i.e., they are not expected to occur in real quantum computers [Blume-Kohout et al., 2022]. Our networks therefore predict the rates of every error from a relatively small set of error types $\mathbb{G} = \{G_1, \ldots, G_k\}$ containing the $k$ most plausible kinds of error. $\mathbb{G}$ is a hyperparameter of our networks. It can be chosen to reflect the known physics of a particular quantum computer and/or optimized using hyperparameter tuning. In our demonstrations, we choose $\mathbb{G}$ to contain all one-body H and S errors as well as all two-body H and S errors that interact pairs of qubits within $h$ steps on the modelled quantum computer's connectivity graph for some constant $h$ (see Fig. 1b-c, where Fig. 1c shows an H or S error in $\mathbb{G}$ if $h \geq 2$). This choice for $\mathbb{G}$ encodes the physical principles that errors are primarily either localized to a qubit or are two-body interactions between nearby qubits [Blume-Kohout et al., 2022]. The size of $\mathbb{G}$ grows with $n$, and for planar connectivity graphs (as in, e.g., contemporary superconducting qubit systems [Arute et al., 2019]) it grows linearly in $n$. This results in $k = \mathcal{O}(n)$ errors whose rates $\mathcal{N}$ must learn to predict.

The internal structures of $\mathcal{N}$ are chosen to reflect general physical principles for how $\mathcal{E}$ and $\vec{m}$ depend on $c$. $\mathcal{E}_{ij}$ is a prediction for the rate that $G_j$ occurs in circuit layer $i$, and this error corresponds to a space/time location within $c$—because it occurs at layer index or time $i$ and $G_j$ acts on a subset of the qubits $Q(G_j)$ (see example in Fig. 1c). This error's rate will therefore primarily depend only on the gates in a time- and space-local region around its location in $c$. Furthermore, this dependence will typically be invariant under time translations (this is true except for some exotic non-Markovian kinds of errors, which we discuss in Section 8.1). We can encode these structures into $\mathcal{N}$ by predicting $\mathcal{E}_{ij}$ from a space-time "window" of $c$ around the associated error's location using a filter $W_j$ that "slides" across the circuit to predict the rate of $G_j$ versus time $i$. Stated more formally, we predict $\mathcal{E}_{ij}$ using a multilayer perceptron $\mathcal{N}_j$ whereby $\mathcal{N}_j(W_j[I(c), i]) = \mathcal{E}_{ij}$ and $W_j[I(c), i]$ is a snippet of $I(c)$ whose temporal origin is $i$ (see Fig. 1e). The shape of each filter $W_j$ is a hyperparameter of our networks and it can be designed to reflect general physical principles, the known physics of a particular quantum computing system, and/or optimized with hyperparameter tuning. The particular neural networks we present later herein use filters $W_j(I(c), i)$ that snip out only layer $i$ and discard the parts of the layer that act on qubits more than $l$ steps away from $Q(G_j)$ in the quantum computer's connectivity graph (e.g., the filter shown in Fig. 1e corresponds to the error shown in Fig. 1c and $l = 1$). This neural network structure has close connections to graph convolution layers [Kipf and Welling, 2016], as well as CNNs. We choose this structure as it can model spatially localized crosstalk errors, which are a ubiquitous but hard-to-model class of errors in quantum computers [Sarovar et al., 2020].

The network $\mathcal{N}$ must also predict the rates of errors that occur during measurements (unless the qpa-NN will only ever predict $F(c)$ not $\mathrm{PST}(c)$), but these are typically independent of the rates of gate errors (which are predicted by the $\mathcal{N}_j$). So we do not use the $\mathcal{N}_j$ and their convolutional filters $W_j$ to make predictions for $\vec{m}$. Instead we use separate but structurally equivalent networks $\mathcal{N}'_j$ with

corresponding filters $W_j'$ that take $M(c)$ as input and implement only spatial filtering. That is, $W_j'$ simply discards rows from $M(c)$, as, unlike $I(c)$, $M(c)$ has no temporal dimension. The $W_j'$ are hyperparameters of our networks allowing us to separately adjust the shape of each $W_j'$ to reflect the known physics of errors induced by measuring qubits. In our demonstrations, our $W_j'$ filters have the same structure as the $W_j$ filters but with an independent $l'$ steps parameter (large $l'$ enables modelling many-qubit measurement crosstalk).

## 3.2 Processing predicted error rates to predict capabilities

We process $\mathcal{N}$'s output to predict $\mathrm{PST}(c)$ or $F(c)$ using a function $f$ with no learnable parameters. This turns $\mathcal{N}$'s output into the two quantities of interest, and it also makes training $\mathcal{N}$ feasible. We cannot easily train $\mathcal{N}$ in isolation because the error matrix $\mathcal{E}$ predicted by $\mathcal{N}$ is not a directly observable quantity. Generating the data needed to train $\mathcal{N}$ directly would require extraordinarily expensive quantum process tomography [Nielsen et al., 2021], which is infeasible except for very small $n$. In contrast, both $\mathrm{PST}(c)$ and $F(c)$ can be efficiently estimated (see Section 2) for a given circuit $c$.

The function $f$ computes an approximation to the value for $\mathrm{PST}(c)$ or $F(c)$ predicted by $\mathcal{E}$ and $\vec{m}$. The matrix $\mathcal{E}$ encodes the prediction that $c$'s imperfect action is

$$\tilde{\mathcal{U}}(c) = \Lambda_d(\mathcal{E})\mathcal{U}(L_d)\cdots\Lambda_1(\mathcal{E})\mathcal{U}(L_1), \tag{6}$$

where the $L_i$ are the $d$ layers of $c$ (see Section 2) and $\Lambda_i(\mathcal{E}) = \exp(\sum_{j=1}^k \mathcal{E}_{ij} G_j)$, i.e., $\Lambda_i(\mathcal{E})$ is an error channel parameterized by the $i^{\text{th}}$ column of $\mathcal{E}$. Equation (6) implies an exact prediction for $\mathrm{PST}(c)$ or $F(c)$ [e.g., Eq. (3)], but exactly computing that prediction involves explicitly creating and multiplying together each of the $4^n \times 4^n$ matrices in Eq. (6). This is infeasible, except for very small $n$. Instead our $f$ computes an efficient approximation to this prediction.

Our function $f$'s action is most easily described by embedding $\mathcal{E}$ into the space of all possible H and S errors $\mathbb{G}_n$, resulting in a $d \times (2^{2n+1} - 2)$ matrix $\mathcal{E}_e$ whose columns are $k$-sparse. However, we never construct these exponentially large matrices. Consider pulling each error channel to the end of the circuit, giving $\tilde{\mathcal{U}}(c) = \Lambda_d'(\mathcal{E}_e')\cdots\Lambda_1'(\mathcal{E}_e')\mathcal{U}(c)$ where $\Lambda_d'(\mathcal{E}_e') = \exp(\sum_{j=1}^{2^{2n+1}-2}[\mathcal{E}_e']_{ij} G_j)$. Because $c$ contains only Clifford gates and Clifford unitaries preserve the Pauli group [Aaronson and Gottesman, 2004], $\mathcal{E}_e'$ has columns that are just $c$-dependent signed permutations of $\mathcal{E}_e$'s columns. The signed permutations required can be efficiently computed in advance (i.e., as an input encoding step) using an efficient representation of Clifford unitaries [Gidney, 2021]. Furthermore, these permutations can be efficiently represented in two $d \times k$ matrices: a *sign matrix* $S(c)$ containing $\pm 1$ signs to be element-wise multiplied with $\mathcal{E}$ and a *permutation indices matrix* $P(c)$ containing integers between 1 and $2^{2n+1} - 2$, where $P_{ij}$ specifies what error $G_j$ becomes when pulled through the $d - i$ circuit layers after layer $i$.

We now have a representation of $\mathcal{E}$'s prediction for the circuit $c$'s error map $\Lambda(c)$ as a sequence of error maps $\Lambda_d'(\mathcal{E}_e')\cdots\Lambda_1'(\mathcal{E}_e')$, and we need to predict $\mathrm{PST}(c)$ or $F(c)$. We can do so if we can compute $\mathcal{E}$'s prediction for the S and H error rates in $\Lambda(c)$, as we can then apply Eq. (4) or Eq. (5). To achieve this, we combine the $\Lambda_i'(\mathcal{E})$ into a single error map using a first-order Baker-Campbell-Hausdorff (BCH) expansion. Using our embedded representation, this means simply approximating $\Lambda(c)$ as $\Lambda(c) \approx \exp(\sum_j v_j G_j')$ where $v_j = \sum_{i=1}^d [\mathcal{E}_e']_{ij}$, i.e., we sum over the rows of $\mathcal{E}_e'$. To predict $F(c)$ we then simply apply Eq. (4) (meaning summing up $v_j$ with those elements that correspond to Hamiltonian errors squared). Because measurement errors impact $\mathrm{PST}(c)$, to predict $\mathrm{PST}(c)$ we again apply the BCH expansion to combine in the predicted measurement error map $\exp(\sum_{j=1}^l m_j G_j)$ and then apply Eq. (5). The efficient representation of the overall action of $f$ is illustrated in Fig. 1 (the addition of the measurement error map is not shown).

## 4 Datasets

### 4.1 Experimental 5-qubit data

We used the 5-qubit datasets from Hothem et al. [2024b] for our experimental demonstrations. Each of these datasets $D = \{(c, \widehat{\mathrm{PST}(c)})\}$ was gathered by running random and periodic mirror circuits

(two types of definite-outcome circuits) on 5-qubit IBM Q computers (`ibmq_london`, `ibmq_essex`, `ibmq_burlington`, `ibmq_vigo`, `ibmq_ourense` and `ibmq_yorktown`), and estimating the PST of each circuit using Eq. (2). Each circuit was run between 1024 and 4096 times, with the exact number depending upon how many times the circuit sampling process generated the circuit (some short, 1-qubit circuits were generated multiple times). The random and periodic mirror circuits contained between 1 and 5 active qubits—called the circuit's *width*—and ranged in depth from 3 to 515 layers (alt. 259 layers for the `ibmq_yorktown` dataset).

As we focus on high-PST circuits, we removed all circuits with a PST less than $85\%$ from each dataset, leaving between 864 (`ibmq_burlington`) and 1369 (`ibmq_yorktown`) circuits in each dataset. The remaining circuits were partitioned into training, validation, and test sets by their original assignment in Hothem et al. [2024b]. This setup enables a direct comparison between our qpa-NNs and the CNNs trained in Hothem et al. [2024b]. Training set sizes ranged from 682 circuits on `ibmq_burlington` to 1097 circuits on `ibmq_yorktown`, with an approximate training, validation, testing split of $80\%$, $10\%$, and $10\%$, respectively.

## 4.2 Simulated 4-qubit data

For our 4-qubit simulations, we generated 5 datasets of 5000 high-fidelity ($F(c) > 85\%$) random circuits, for a hypothetical 4-qubit processor with a "ring" geometry (i.e., like that in Fig. 1b). The circuits ranged in width ($w$) from 1 to 4 qubits, and in depth from 1 to 180 circuit layers. We designed each circuit for a randomly chosen subset of $w$ qubits. Each circuit layer was created by *i.i.d.* sampling from all possible circuit layers on the $w$ active qubits. We used a gate set containing two-qubit CNOT gates and 7 different single-qubit gates (specifically $\{X(\pi/2), Y(\pi/2), X(3\pi/2), Y(3\pi/2), X(\pi), Y(\pi), Z(\pi)\}$ where $P(\theta)$ denotes a rotation around the $P$ axis of the Bloch sphere by $\theta$). See Appendix C for additional details.

All circuits were simulated under the same error model, consisting of local coherent (i.e., H) errors, to exactly compute each $c$'s $F(c)$ [Fig. 3 shows a histogram of $F(c)$]. After removing duplicate circuits, the resulting datasets $D = \{(c, F(c))\}$ were partitioned into training, validation, and testing subsets, with a partition of $56.25\%$, $18.75\%$, and $25\%$, respectively. The parameters of the error model were randomly selected: each gate was assigned a small error strength, which was then distributed randomly across all possible (local) one- or two-qubit coherent errors, for the one- and two-qubit gates, respectively. We chose a model with only coherent errors as these errors are ubiquitous, they are hard to model accurately and efficiently, and we conjecture that qpa-NNs can model them.

We also generated 5 datasets of 750 random mirror circuits on the same hypothetical 4-qubit quantum computer. Again, the random mirror circuits varied in width from 1 to 4 qubits, and were designed to be run on a randomly selected subset of $w$ qubits. However, instead of *i.i.d.* sampling of each circuit layer, each circuit was randomly sampled from the class of random mirror circuits on the $w$ qubits. The depth of the mirror circuits ranged from 8 to 174 layers. Because we generated the mirror circuit datasets to evaluate how well qpa-NNs and CNNs generalize to out-of-distribution circuits, they were used exclusively as testing sets. To ensure that no training was performed on mirror circuits, we removed any mirror circuits that appeared in the random circuit sets (in actuality, there were no duplicates).

## 4.3 Simulated 100-qubit data

For our 100-qubit simulation, we generated a single dataset of 5000 high-fidelity ($F(c) > 91\%$) random circuits, for a hypothetical 100-qubit quantum computer with a "ring" geometry. All of the circuits had a width of 100 qubits, and ranged in depth from 1 to 22 circuit layers. We sampled circuit layers using the same process and gate set as in the 4-qubit simulations.

We simulated every circuit using the same error model, consisting of local, weight-1 S and H errors. As before, the parameters of the error model were randomly selected and the data were partitioned into training, validation, and testing subsets according to a $56.25\%$, $18.75\%$, $25\%$ split.

Unlike in our 4-qubit simulations, we did not compute $F(c)$ exactly as doing so for a 100-qubit circuit is infeasible in the presence of coherent errors. Instead, we used a first-order simulation method to approximate $F(c)$. In this method, $F(c)$ is computed by assigning each gate its own error vector based on the error model, adding up the error vectors layer-wise to compute an error vector

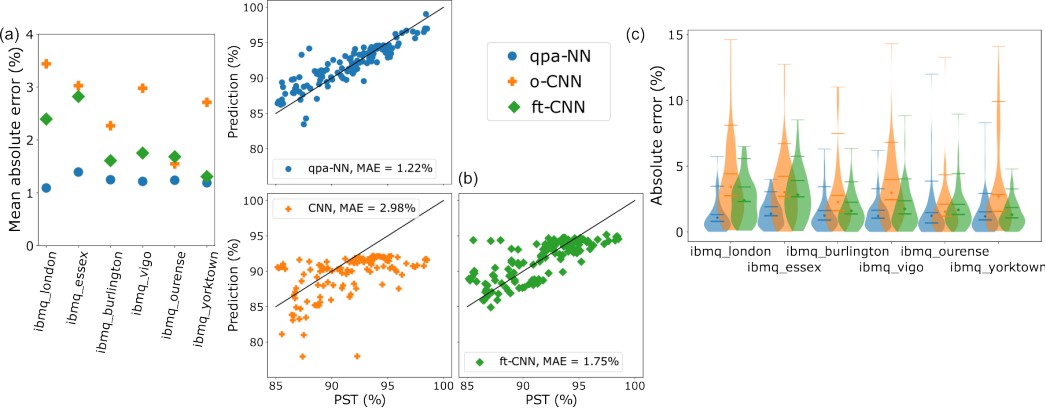

Figure 2: **Prediction accuracy on real quantum computers. (a)** The mean absolute error of our qpa-NNs (●), the CNNs from Hothem et al. [2024b] (o-CNN, +), and fine-tuned CNNs (ft-CNN, ◆) on the test data. **(b)** The predictions of the three models for `ibmq_vigo` on the test data, and **(c)** the distribution of each model's absolute error on the test data, including the 50th, 75th, 95th and 100th percentiles (lines) and the means (points).

for each circuit layer, and then computing $F(c)$ as in the second part of a qpa-NN [Figure 1(f)]. See Appendix C.4 for more details.

### 4.4 Encoding schemes

We used two different encoding schemes for converting each circuit $c$ into a tensor. For the CNNs on experimental data, we used the same encoding scheme as Hothem et al. [2024b], as we used their data and networks. For all qpa-NNs, and the CNNs on simulated data, we used the following scheme. As outlined in Section 3, each width-$w$ circuit $c$ is represented by a three-dimensional tensor $I(c) \in \{0,1\}^{n \times d(c) \times n_{ch}}$ describing the gates in $c$ and a matrix $M(C) \in \{0,1\}^{2 \times w}$ describing the measurement of the qubits. The $ij$-th entry of $I(c)$,

$$I_{ij}(c) = (I_{ij1}(c), \ldots, I_{ijn_{ch}}(c)), \tag{7}$$

is a one-hot encoded vector of what happens to qubit $i$ in layer $j$. For the hypothetical 4-qubit ring processor, $n_{ch} = 11$: one channel for each single-qubit gate and four channels for the CNOT gates. There are four CNOT channels to specify if the qubit $i$ is the target or control qubit and if the interacting qubit is to the left or right of qubit $i$. We used an additional 4 or 8 CNOT channels for the experimental data, depending on the quantum computer's geometry. The first row in $M(c)$ is the bitstring specifying which qubits are measured at the end of $c$. When $c$ is a definite-outcome circuit, the second row is its target bit string, i.e., the sole bit string in the support of $c$'s outcome distribution when it is executed without error [i.e., P($c$)]. Both $I(c)$ and $M(c)$ are zero-padded to ensure a consistent tensor shape across a dataset.

Additionally, each circuit $c$ is accompanied by a permutation matrix $P(c) \in \mathbb{N}^{n \times k}$ and sign matrix $S(c) \in \{\pm 1\}^{n \times k}$. The $ij$-entry of $P(c)$ specifies which error the $j$-th tracked error occurring after the $i$-th layer is transformed into at the end of the circuit. The $ij$-th entry of $S(c)$ specifies the sign of that error.

## 5  5-qubit experiments

We now present the results from our head-to-head comparison between the qpa-NNs and the CNNs on the 5-qubit datasets used in Hothem et al. [2024b]. Figure 2 shows the mean absolute error (MAE) achieved by the CNNs (+) and the qpa-NNs (●) on each of the datasets. For all datasets, MAE is lower for the qpa-NNs than the CNNs, with an average reduction of $50.4\%$ ($\sigma_x = 16.7\%$, i.e., the standard deviation of percent-drop in MAE). The Bayes factor $K$ is between $K = 10^{30}$ and $K = 10^{383}$ (here, $K$ is the ratio of the likelihood of the qpa-NN to the likelihood of the CNN given the test data). This is overwhelming evidence that the qpa-NN is a better model ($K \geq 10^2$ is typically

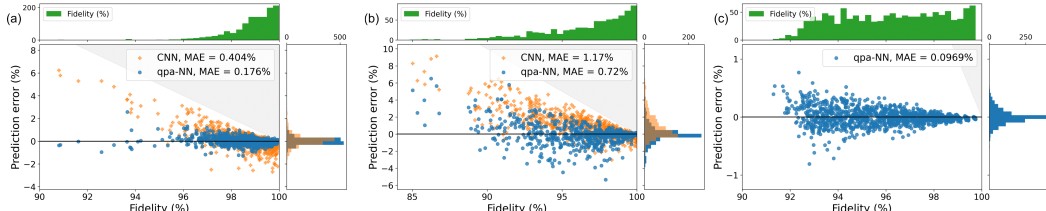

Figure 3: **Demonstrating our qpa-NNs' accuracy for hard-to-model coherent errors and at scale. (a)** Scatter plot of the prediction errors on test data of a qpa-NN (●) and CNN (●) trained to predict the fidelity $F(c)$ of random circuits run on a hypothetical 4-qubit quantum computer. The qpa-NN significantly outperforms the CNN. The top subplot contains a histogram (green bars) of the ground-truth fidelities. **(b)** Prediction errors on out-of-distribution test data, from random mirror circuits. The qpa-NN achieves modest prediction accuracy on this out-of-distribution task, suggesting that the qpa-NNs are accurately learning error rates. **(c)** Prediction errors on the 100-qubit test data, demonstrating that our qpa-NN approach can accurately predict $F(c)$ for circuits run on large-scale quantum computers.

considered decisive). These results strongly suggest that the extra infrastructure in the qpa-NNs is making a difference.

The improved performance of the qpa-NNs is not because of an increase in model size. For example, the `ibmq_london` CNN contains $6,649,531$ trainable parameters compared to the $1,218,348$ trainable parameters in the qpa-NN. Moreover, CNNs of similar or larger sizes than the qpa-NNs were included in the hyperparameter optimization space of the CNNs [Hothem et al., 2024b].

Nonetheless, comparing the qpa-NNs to the CNNs is somewhat unfair as the CNNs were trained on out-of-distribution circuits—they were trained on the entire training dataset from Hothem et al. [2024b] which also contains low-PST circuits. For a fairer comparison, we fine-tuned each CNN (Fig. 2, ◆) on the same high-PST training set used to train the qpa-NNs. Fine-tuning typically increased the CNNs' performances (mean $25.1\%$ improvement, $\sigma_x = 22.3\%$). However, the qpa-NNs achieve a MAE that is lower than the fine-tuned CNNs by $32.2\%$ on average ($\sigma_x = 17.3\%$) and outperform the fine-tuned CNNs on all six datasets. $K$ is between $10^{28}$ and $10^{238}$, which is overwhelming evidence that the qpa-NNs are better models than the fine-tuned CNNs.

## 6   4-qubit simulations

One reason why the extra infrastructure in our qpa-NNs may be necessary is that off-the-shelf networks struggle with modeling coherent errors [Hothem et al., 2024b]. To test our hypothesis, we trained a qpa-NN to predict the fidelity $F(c)$ of random circuits executed on a hypothetical 4-qubit quantum computer experiencing purely coherent errors. We compared this qpa-NN to a hyperparameter-tuned CNN trained on the same data. Figure 3 shows the results from one representative dataset.

The qpa-NNs again significantly outperform the CNNs. Across the five datasets, the qpa-NNs' averaged a $52.4\%$ reduction in MAE ($\sigma_x = 3.00\%$) on the test data. We also see a significant improvement in the mean Pearson correlation coefficient, $\bar{r}_{\text{qpa-NN}} = .968$ vs. $\bar{r}_{\text{CNN}} = .749$.

We also found that qpa-NNs trained on random circuits are modest predictors of the infidelity of random mirror circuits, which are a different family of circuits. This is an example of out-of-distribution generalization. Random mirror circuits differ in a variety of ways from the random circuits on which the qpa-NNs were trained, including both the presence of idle gates (which are noiseless in our simulations) and a motion-reversal structure in the circuits that causes the addition or cancellation of errors that are far apart in time. The qpa-NNs achieve an average MAE of .72% on the random mirror circuits ($\sigma_x = .046\%$). Although this is a $3.2\times$ increase in MAE over the in-distribution test data, the strong linear relation between the network's predictions and the ground truth ($\bar{r} = .912$, $\sigma_x = .009$) strongly suggests that the qpa-NNs are learning information relevant to random mirror circuits.

# 7  100-qubit simulation

Quantum-physics-aware neural networks scale just as well as CNNs, despite their extra infrastructure. To demonstrate their scalability, we trained a qpa-NN to predict the fidelity $F(c)$ of random circuits executed on a hypothetical 100-qubit quantum computer experiencing a mix of stochastic and coherent errors. To our knowledge, this is the first creation of a capability model of any kind, for a 100+ qubit quantum computer. Figure 3(c) shows the results from our demonstration.

The qpa-NN achieved a MAE of $0.097\%$. While the underlying noise model was quite simple, this result shows that it is technically feasible to construct qpa-NN capability models for today's moderate-scale quantum computers and for tomorrow's early fault-tolerant quantum computers.

# 8  Discussion

## 8.1  Limitations

Our results are a significant improvement over the state of the art, but our approach does have several limitations:

1. As presently conceived, our approach assumes that the modelled quantum computer's error rates are invariant under time translations, which is a kind of Markovianity assumption (although it is weaker than the typical Markovian assumption used in conventional quantum computer models [Nielsen et al., 2021]). However, non-Markovian noise exists in quantum computers [White et al., 2020]. In the future, we plan to address this issue by adding temporal information into our approach, perhaps with a temporal or positional encoding [Vaswani et al., 2017].

2. Our approach only considers two error classes (H and S errors). Other Markovian error classes, like amplitude damping, exist, but their error rates $\varepsilon$ contribute to PST and fidelity at order $\mathcal{O}(\varepsilon^3)$ [Mądzik et al., 2022]. Our approach can be easily extended to include those errors, if necessary, by learning their rates with $\mathcal{N}$ and updating $f$ to account for their presence.

3. Our current approach works for Clifford circuits, which includes arguably the most important kinds of circuits (e.g., quantum error correction circuits) but not all interesting circuits. This is because our method for efficiently propagating errors through circuits (implemented by $f$ together with the $S$ and $P$ matrices) leverages the elegant mathematics of Clifford circuits. Our approach can be easily extended to generic few-qubit quantum circuits ($\lesssim 10$ qubits), but to obtain the efficiency needed for large $n$ with general circuits we will need to develop approximate methods for propagating errors through those circuits.

## 8.2  Conclusion

In this paper, we presented a new quantum-physics-aware neural network architecture for modelling a quantum computer's capability that significantly improves upon the state of the art. The new architecture concatenates two parts: (i) a neural network with structural similarities to GNNs that uses gate information and a quantum computer's connectivity graph to predict the rates of errors in each of a circuit's layers, and (ii) a non-trainable function that turns the predicted error rates into a capability prediction. By imbuing these networks with knowledge about how errors occur and combine within a circuit, we are able to outperform state-of-the-art CNN-based capability models by $\sim 50\%$ on both experimental data and simulated data. We also provided evidence that our quantum-physics-aware networks are learning the true physical error rates, as they exhibit modest prediction accuracy when predicting the fidelity of out-of-distribution quantum circuits, which would enable our networks to also be used to diagnose the error processes occurring in a particular quantum computer (an important task known as characterization or tomography [Nielsen et al., 2021]).

Understanding which quantum circuits a quantum computer can run, and how well it can run them, is an important yet challenging component of understanding a quantum computer's power. Given the complexity of the problem, neural networks are likely to play a large role in its solution. As our results demonstrate, our new physics-aware network architecture could play a critical role in building fast and reliable neural network-based capability models.

## Acknowledgments and Disclosure of Funding

This material was funded in part by the U.S. Department of Energy, Office of Science, Office of Advanced Scientific Computing Research, Quantum Testbed Pathfinder Program, and by the Laboratory Directed Research and Development program at Sandia National Laboratories. T.P. acknowledges support from an Office of Advanced Scientific Computing Research Early Career Award. We acknowledge the use of IBM Quantum services for this work. The views expressed are those of the authors, and do not reflect the official policy or position of IBM or the IBM Quantum team.

Sandia National Laboratories is a multi-mission laboratory managed and operated by National Technology & Engineering Solutions of Sandia, LLC (NTESS), a wholly owned subsidiary of Honeywell International Inc., for the U.S. Department of Energy's National Nuclear Security Administration (DOE/NNSA) under contract DE-NA0003525. This written work is authored by an employee of NTESS. The employee, not NTESS, owns the right, title and interest in and to the written work and is responsible for its contents. Any subjective views or opinions that might be expressed in the written work do not necessarily represent the views of the U.S. Government. The publisher acknowledges that the U.S. Government retains a non-exclusive, paid-up, irrevocable, world-wide license to publish or reproduce the published form of this written work or allow others to do so, for U.S. Government purposes. The DOE will provide public access to results of federally sponsored research in accordance with the DOE Public Access Plan.

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

## A  Compute resources

All of the quantum-physics-aware neural networks used in our 5-qubit experiments and 4-qubit simulations were trained using a 6-Core Intel Core i9 processor on a MacBookPro 15.1 with 32GB of memory. Each model took roughly 15-20 wall clock minutes to train. Total training time, across the paper, totaled $\sim 160$ wall clock minutes.

All of the 4-qubit simulations and data pre-processing were performed using a 6-core Intel Core i9 processor on a MacBookPro 15.1 with 32GB of memory. Each dataset took approximately 1 hour of wall clock time to create. This total includes the initial circuit creation, simulating the circuits, and encoding each circuit into a tensor.

All of the 100-qubit simulations, data pre-processing, and model training were performed using two 14-core Intel Xeon CPU E5-2697 v3 @ 2.60GHz processors. An end-to-end run (i.e., circuit generation to trained model predictions) took roughly 12 hours of wall clock time.

## B  Code and data availability

The simulated data as well as records of all the quantum physics-aware networks will be released publicly once they clear Sandia's copyright process. Until then, please email the authors. The CNNs and 5-qubit experimental datasets used in Hothem et al. [2024b] are available at Hothem et al. [2023a]. The datasets were originally located at Proctor et al. [2021b]. Each dataset was released under a CC-BY 4.0 International license.

All simulations were performed using a combination of `pygsti` version 0.9.11.2 [Nielsen et al., 2020] and `stim` version 1.13.0 [Gidney, 2021]. Models were trained and developed using `Keras` version 2.12.0 [Chollet et al., 2015] and `TensorFlow` version 2.12.0 [Abadi et al., 2015]. The physics-aware network model classes (`CircuitErrorVecScreenZErrorsWithMeasurementsBitstrings` for PST and `CircuitErrorVec` for process fidelity) are available in the Supplementary Material [Hothem et al., 2024a] as well as on the `feature − ml` branch of `pygsti`.

## C  Datasets

| Device | Geometry | Circuit types | Circuit widths | Circuit depths | Training set size | Validation set size | Test set size |
|---|---|---|---|---|---|---|---|
| ibmq_london | t-bar | mirror | 1-5 qubits | 3-515 layers | 711 circuits | 104 circuits | 91 circuits |
| ibmq_ourense | t-bar | mirror | 1-5 qubits | 3-515 layers | 930 circuits | 124 circuits | 114 circuits |
| ibmq_essex | t-bar | mirror | 1-5 qubits | 3-515 layers | 713 circuits | 93 circuits | 86 circuits |
| ibmq_burlington | t-bar | mirror | 1-5 qubits | 3-515 layers | 682 circuits | 90 circuits | 92 circuits |
| ibmq_vigo | t-bar | mirror | 1-5 qubits | 3-515 layers | 1029 circuits | 137 circuits | 126 circuits |
| ibmq_yorktown | bowtie | mirror | 1-5 qubits | 3-515 layers | 1097 circuits | 132 circuits | 140 circuits |
| Ring (x5) | ring | random *i.i.d.* | 1-4 qubits | 1-180 layers | 2813 circuits | 938 circuits | 1250 circuits |
| Ring (x5) | ring | mirror | 1-4 qubits | 8-174 layers | - | - | 750 circuits |
| 100-qubit Ring | ring | random *i.i.d.* | 100 qubits | 1-22 layers | 2812 circuits | 938 circuits | 1250 circuits |

Table 1: **Summary data of every dataset used in the paper.** The data for the 4-qubit ring processors is averaged over the 5 simulated datasets. See Figure 4 for images of each processor geometry (i.e., the qubit connectivity graph).

We provide additional details on the datasets used in the paper. Table 1 summarizes each dataset. We tracked all weight-2 errors with support on qubits connected by 2 hops in all, but the 100-qubit datasets. Below, we provide additional details on the circuit and error model generating processes.

## C.1 Creating the circuits

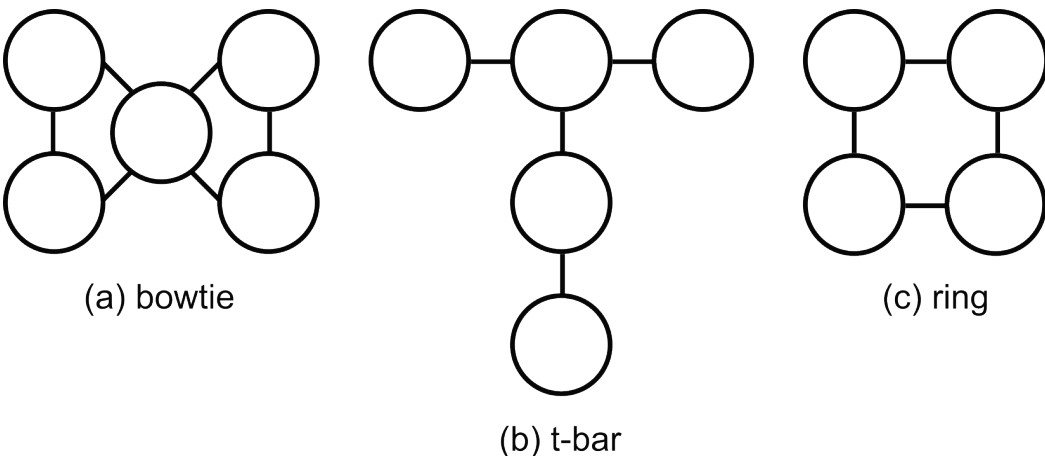

Figure 4: **Device geometries.** The connectivity graphs for the **(a)** 5-qubit `ibmq_yorktown` "bowtie" processor; **(b)** the remaining 5-qubit experimental "t-bar" processors; and **(c)** the 4-qubit simulated "ring" processor. The 100-qubit simulated "ring" processor has the same topology, just more qubits.

In this subsection, we go over how the random *i.i.d.*-layer circuits and random mirror circuits were created for this paper. We start by explaining how we generated the random *i.i.d.*-layer circuits for a 4-qubit ring processor, and then explain the modifications needed to generate the random mirror circuits. This subsection's content is conceptual. The actual circuits were created in `pygsti` using the code in the Supplementary Material.

Each random *i.i.d.*-layer circuit $c$ was created by a multi-step process. First, we randomly sampled a connected subset $\mathbb{Q}_c \subseteq \{Q0, Q1, Q2, Q3\}$ of qubits for which $c$ is designed for. Then, we uniformly sampled $c$'s depth from between 1 and $d_w$, a pre-determined, circuit-width-dependent maximum depth. The depths $d_w$ were selected to ensure that $F(c) > 85\%$ given the maximum error strengths used to create the simulated error model (Section C.2). Third, we randomly sampled a two-qubit gate density $\rho_{2Q}$ between 0 and $2/3$. The density $\rho_{2Q}$ determines the average number of two-qubit gates in each of $c$'s layers. We then sampled each layer *i.i.d.* from all possible circuit layers on the qubits in $\mathbb{Q}_c$.

The random mirror circuits were generated using a similar multi-step process with two differences. The first difference is that we used a pre-determined maximum depth of $d_w/6$. We chose to reduce the pre-determined, circuit-width-dependent maximum depth so that the deepest random mirror circuits had roughly the same length as the deepest random *i.i.d.* circuits. The second difference is that we created a random mirror circuit on $\mathbb{Q}_c$. See Proctor et al. [2021a] for more details.

## C.2 Creating a 4-qubit error model

In this subsection, we explain how we constructed the 4-qubit Markovian local coherent error model used in Section 6. Again, we provide a conceptual explanation. The actual error model was created in `pygsti` using the code found in the Supplementary Material.

The 4-qubit Markovian local coherent error model was specified using the error generator framework explained in Section 2 and Blume-Kohout et al. [2022]. The error model consists of operation-dependent errors sampled according to a two-step process. The error strengths for each gate and qubit(s) pairs were independently sampled. First, we sampled an overall error strength $\varepsilon_g$ for each one- and two-qubit gate $g$ by randomly sampling from $[0, 1]$ and scaling by a pre-determined maximum error strength $(.025\%)$. Then we sampled the relative error strengths $\vec{\varepsilon}_{g,\text{rel}}$ of each of the $4^n - 1$ coherent errors, where $n = 1, 2$ for one- and two-qubit gates, respectively. We then normalized $\vec{\varepsilon}_{g,\text{rel}}$

to obtain the actual error strengths according to the following equation:

$$\vec{\varepsilon}_g = \frac{\sqrt{\varepsilon_g} \cdot \vec{\varepsilon}_{g,\mathrm{rel}}}{\sqrt{\sum_i \varepsilon_{g,i}^2}}.$$ (8)

The re-scaling ensures that, to first order, gate $g$ contributes approximately $\varepsilon_g$ to the circuit's process infidelity (or PST, if appropriate).

### C.3 Creating a 100-qubit error model

In this subsection, we explain how we constructed the 100-qubit Markovian error model used in Section 7. As with the 4-qubit error model, we provide a conceptual explanation. The actual error model was created in `pygsti`.

As with the 4-qubit error model, the 100-qubit Markovian error model was specified using the error generator framework explained in Section 2. The error model consisted of gate-dependent errors sampled according to a two-step process. Unlike with the 4-qubit error model, the 100-qubit error model included non-local Pauli stochastic and coherent errors, and the errors for each gate were independent of the qubit acted upon by the gate. Moreover, all of the errors in the 100-qubit error model are weight-1 errors (i.e., they affect a single qubit).

Each gate's error strengths were independently sampled. First, we enumerated all 600 possible weight-1 Pauli stochastic and coherent errors in a 100-qubit device. Then, for each gate $g$, we independently sampled the strength of each of the 300 weight-1 Pauli stochastic errors and the 300 possible weight-1 coherent errors. The strengths were sampled uniformly random, with a maximum Pauli stochastic error strength of 0.0000001 and a maximum coherent error strength of 0.00005. The resulting 600 error strengths were assembled into a 600-dimensional error vector for the gate, $\vec{\varepsilon}_g$.

### C.4 Simulating the 100-qubit circuits

In this subsection, we describe the first-order simulation method used to approximate $F(c)$ in our 100-qubit simulations (Section 4.3). The method works by constructing an approximate error matrix $E(c)$ [Fig. 1(e)], and then estimating $F(c)$ by performing the same computation as in the second half of a qpa-NN [Fig. 1(f)]. Herein, we describe how to construct the approximate error matrix $E(c)$. Readers should refer to Section 3 for an in-depth explanation on how to use $E(c)$ to estimate $F(c)$.

For each circuit $c$, we constructed an approximate error matrix $E(c)$ by concatenating approximate error *vectors* $E_i(c)$ for each circuit layer in $c$. Each layer's error vector was computed as a linear combination of the individual gate error vectors, with coefficients equal to the number of times each gate appears in the circuit layer.

## D Networks

### D.1 Quantum-physics-aware network details

| Dataset | Metric | Model size | $N_{\mathrm{hops}}$ | $N_{\mathrm{errors}}$ | Dense units |
|---------|--------|-----------|--------|---------|-------------|
| 5-qubit t-bar | $\mathrm{PST}(c)$ | 1218348 | 3 | 174 | [30, 20, 10, 5, 5, 1] |
| 5-qubit bowtie | $\mathrm{PST}(c)$ | 1596420 | 3 | 210 | [30, 20, 10, 5, 5, 1] |
| 4-qubit ring | $F(c)$ | 299772 | 2 | 132 | [30, 20, 10, 5, 5, 1] |
| 100-qubit ring | $F(c)$ | 12706200 | 1 | 600 | [30, 20, 10, 5, 5, 1] |

Table 2: **Summary data for the quantum-physics-aware networks used in the paper.**

Table 2 briefly outlines the hyperparameters and model sizes of the physics-aware neural networks used in this paper. The $N_{\mathrm{hops}}$ and $N_{\mathrm{errors}}$ hyperparameters were chosen by hand based upon subject-matter-expert knowledge of the errors in a quantum computer. The size and shape of the dense layers

were selected arbitrarily. All dense subunits used a ReLU activation function. All models were trained using keras's Adam [Kingma and Ba, 2015] optimizer with a step size of $10^{-3}$ and with mean squared error as the loss function. Model training was cut short using early stopping. To help with training, we scaled $\text{PST}(c)$ and $F(c)$ by 10000 when training the physics-aware networks. The notebooks in the Supplementary Material contain more details.

### D.2 Convolutional neural network details

Details on the specific convolutional neural networks used in this paper are located in Hothem et al. [2024b]. We fine-tuned each network on high-PST experimental data using the Adam optimizer and early stopping.

We selected the model architecture for the convolutional neural networks used in our simulations via hyperparameter tuning. We performed 100 trials of hyperparameter tuning using the BayesianOptimization class in kerastuner. Additional details, including the specific hyperparameter space, are located in the Supplementary Material.

## E    Experimental results

| Dataset | Network | Mean absolute error (%) | Bayes factor vs. CNN | Bayes factor vs. ft-CNN |
|---|---|---|---|---|
| | qpa-NN | 1.09 | $10^{340}$ | $10^{184}$ |
| ibmq_london | CNN | 3.44 | - | - |
| | ft-CNN | 2.39 | $10^{156}$ | - |
| | qpa-NN | 1.24 | $10^{30}$ | $10^{33}$ |
| ibmq_ourense | CNN | 1.55 | - | - |
| | ft-CNN | 1.68 | $10^{-2.48}$ | - |
| | qpa-NN | 1.39 | $10^{248}$ | $10^{238}$ |
| ibmq_essex | CNN | 3.03 | - | - |
| | ft-CNN | 2.82 | $10^{9.46}$ | - |
| | qpa-NN | 1.25 | $10^{152}$ | $10^{58.2}$ |
| ibmq_burlington | CNN | 2.27 | - | - |
| | ft-CNN | 1.61 | $10^{93.7}$ | |
| | qpa-NN | 1.21 | $10^{378}$ | $10^{115}$ |
| ibmq_vigo | CNN | 2.98 | - | - |
| | ft-CNN | 1.75 | $10^{263}$ | - |
| | qpa-NN | 1.19 | $10^{383}$ | $10^{28.4}$ |
| ibmq_yorktown | CNN | 2.71 | - | - |
| | ft-CNN | 1.31 | $10^{354}$ | - |

Table 3: **Summary model performance data on experimental data.**

Table 3 summarizes model performance on each of the experimental datasets used in the paper. Copies of the pre-trained quantum-physics-aware neural networks, original CNNs, and fine-tuned CNNs are available in the Supplementary Material. Scatter plots of each model's predictions are also available in the Supplementary Material.

## F    Simulation results

Table 4 summarizes model performance on each of the 4-qubit simulated datasets used in the paper. Copies of the pre-trained quantum-physics-aware neural networks and fine-tuned CNNs are available in the Supplementary Material. Scatter plots of each model's prediction errors are also available in the Supplementary Material.

| Dataset | Network | Circuit type | Mean absolute error (%) | Pearson correlation coefficient |
|---|---|---|---|---|
| 0 | qpa-NN | random *i.i.d.* | .176 | .968 |
| | | mirror | .720 | .914 |
| | CNN | random *i.i.d.* | .404 | .751 |
| | | mirror | 1.17 | .872 |
| 1 | qpa-NN | random *i.i.d.* | .200 | .960 |
| | | mirror | .652 | .921 |
| | CNN | random *i.i.d.* | .421 | .741 |
| | | mirror | .922 | .861 |
| 2 | qpa-NN | random *i.i.d.* | .190 | .970 |
| | | mirror | .769 | .914 |
| | CNN | random *i.i.d.* | .406 | .732 |
| | | mirror | 1.20 | .853 |
| 3 | qpa-NN | random *i.i.d.* | .191 | .952 |
| | | mirror | .719 | .912 |
| | CNN | random *i.i.d.* | .367 | .764 |
| | | mirror | 1.02 | .857 |
| 4 | qpa-NN | random *i.i.d.* | .195 | .960 |
| | | mirror | .761 | .898 |
| | CNN | random *i.i.d.* | .405 | .763 |
| | | mirror | 1.05 | .847 |

Table 4: **Summary model performance data on the 4-qubit simulated data.**

