# OpenReview forum: "What is my quantum computer good for? Quantum capability learning with physics-aware neural networks"
_NeurIPS.cc/2024/Conference — NeurIPS 2024 poster_

### Official Review · Reviewer_k1xb · 2024-07-06

**Soundness:** 3
**Presentation:** 3
**Contribution:** 2
**Rating:** 6
**Confidence:** 3

**Summary:**

The paper introduces a novel quantum-physics-aware neural network (qpa-NN) architecture for quantum capability learning. The model achieves error reduction in capability prediction on both experimental and simulated data.

**Strengths:**

1. The qpa-NN architecture incorporates quantum physics principles, which provides a new perspective of designing models in the field of learning-based quantum capability learning.

2. The demonstrated reduction in mean absolute error over CNN-based models is commendable.

**Weaknesses:**

1. The reviewer's main concern is regarding the qubit scale of the dataset. However, the dataset used in the experiments, which consists of maximum 5 qubits, appears too small both from the perspective of current quantum hardware and classical simulations. The reviewers are curious to know the reason behind the inability to collect data on systems with more qubits (either experimental data or simulated data). Is it due to the high difficulty of experimental deployment, or is it because the current models struggle to train on larger datasets?

2. The evaluation method mentioned in the paper, Process Fidelity (Eq. 3), may have scalability issues. Although the authors attempted to explain in Section 3.2 how some approximations can be used to calculate Eq. 3 relatively efficiently, this description is hard to follow. For instance, can the proposed approximation method avoid exponential computational and storage complexity? Additionally, the reviewer hope that in the revised version, the authors can provide a discussion on the differences between the approximations designed in this paper and the estimation methods in [Proctor et al., 2022], or if the approximations in this paper are merely direct adaptations of the latter's estimation methods.

**Questions:**

How does the proposed architecture handle scalability challenges as the number of qubits increases?

**Limitations:**

The authors have discussed the limitations.

---

> ### Author Rebuttal · Authors · 2024-08-07
>
> We appreciate the reviewer’s summary of our paper and their recognition of its strengths. Your thorough review and constructive feedback have been invaluable in refining our manuscript. Incorporating your suggestions will greatly enhance its quality and impact.
>
> In response to your feedback, we plan to include the following revisions in our final manuscript:
>    - A new large-scale demonstration in which we train a qpa-NN to predict the process fidelity of 100-qubit circuits executed on a simulated quantum computer experiencing low-levels of coherent and stochastic errors. We believe that this demonstration conclusively shows that the qpa-NNs scale to large-scale systems even when predicting process fidelity.
>    - A short discussion on how our qpa-NN approach to capability learning differs from, yet is synergistic with, the mirror circuit fidelity estimation (MCFE) protocol introduced in [Proctor et. al., 2022].
>    - Clearer exposition on how our qpa-NNs estimate process fidelity without exponential and computational storage complexity by only predicting the most influential terms in the first-order approximation to process fidelity (Eq. 3).
>
> We now address, in order, the three listed weaknesses and explain how our new results/edits address each of the weaknesses.
>
> **Weakness 1**: the lack of a large-scale demonstration (same as the response to reviewer beTU).
>
> We agree with the reviewer that the lack of a large-scale demonstration is the primary weakness of our paper, and we hope that our new 100-qubit demonstration will satisfy the reviewer. We believe that the qpa-NNs’ strong performance on this data conclusively demonstrates the scalability of our approach.
>
> Fig. 1 in the rebuttal material depicts the results from training a qpa-NN to predict the process fidelity of circuits run on a noisy 100-qubit quantum computer experiencing low-levels of weight-1 coherent and stochastic errors. Because of the infeasibility of performing a strong simulation of coherent errors on a 100-qubit device (fully modelling coherent errors is equivalent to universal quantum computation), we used a first-order approximation to the strong simulation, like the simulation technique used in the non-Markovian simulations in [Hothem et. al., 2023]. The qpa-NN achieves a mean absolute error of .097%. Crucially, these results demonstrate that is both possible to construct qpa-NNs for arbitrarily large system sizes, and that the qpa-NNs can achieve excellent prediction accuracy at arbitrarily large system sizes.
>
> **Weakness 2**: unclear explanation of scalability.
>
> In addition to including two new empirical demonstrations of our the qpa-NNs’ scalability, we plan to clarify the theoretical arguments for the scalability of qpa-NN’s in the text. There are two primary criteria for developing a scalable machine learning technique to assess the performance of a quantum computer. The first is the ability to efficiently (i.e., without exponential time or sample complexity) gather training data and the second is ensuring that the model’s size grows polynomially in the device size. The qpa-NN approach satisfies both criteria.
>
> We can efficiently gather training data for qpa-NNs trained on experimental hardware, regardless of if we are using probability of successful trials (PST) or process fidelity. A definite-outcome circuit’s PST can be efficiently estimated to 1/sqrt(N) precision by running the circuit N times on the quantum computer, while any circuit’s process fidelity can be estimated to O(1/sqrt(N)) precision by running three ensembles of closely related circuits O(N) times using MCFE [Proctor et. al., 2022]. The only limiting factor to gathering training data on real hardware are the noise levels in current real-world, large-scale devices.
>
> Likewise, we limit the growth of our qpa-NNs’ parameter counts to be polynomial in the number of qubits by only tracking the effect of the most important errors in a device, the local, low-weight errors. Physics intuition tells us that if a gate G acts on qubit Q, then the most probable errors induced by G will affect a local neighborhood of Q. Moreover, these errors are unlikely to affect too many qubits at once. In other words, gates induce low-weight, local errors, as determined by a device’s connectivity graph. Our qpa-NNs reflect this intuition by only trying to predict the impact of local, low-weight errors on a circuit’s fidelity and completely ignoring the contributions of the highly improbable high-weight and non-local errors.
>
> As a result, a qpa-NN’s parameter count grows polynomially in device size. See Fig. 4 for a visualization of how a qpa-NN’s parameter count grows with device size for a fixed set of hyperparameters. For a device with a grid connectivity graph, the parameter count grows quadratically, while it grows linearly for a device with a ring connectivity graph.
>
> **Weakness 3**: lack of comparison with mirror circuit fidelity estimation (MCFE).
>
> We thank the reviewer for highlighting this point of confusion. It appears that we did not clearly explain how our method differs from the method of mirror circuit fidelity estimation (MCFE) proposed in [Proctor et. al., 2022]. We will clarify how our method for training qpa-NNs to predict process fidelity scales and differs from MCFE.
> MCFE and qpa-NNs solve different problems. qpa-NNs aim to construct predictive models of a quantum computer’s capability, predicting the process fidelity of any new circuit from a class after training on a representative sample. MCFE, however, estimates the process fidelity of a known target circuit by running related circuits on a quantum computer. While MCFE does not provide a model, it is useful for gathering training data to build predictive models.

---

> > ### Comment · Reviewer_k1xb · 2024-08-13
> >
> > Thanks for the response. Most of my questions are solved. I would happy to raise my score.

---

### Official Review · Reviewer_beTU · 2024-07-10

**Soundness:** 3
**Presentation:** 3
**Contribution:** 2
**Rating:** 5
**Confidence:** 3

**Summary:**

This paper introduces a neural-network-based architecture for quantum capability learning. The idea is to utilize the architecture to predict rates of errors in quantum circuits. The authors compared their qpa-NN with previous CNN-based method and elucidated their improved performance.

**Strengths:**

1. qpa-NN outperforms the previous CNN method in predicting circuit success probability.

**Weaknesses:**

1. Given that neural network-based methods for predicting circuit success probability have been previously discussed in the literature, the novelty of the approach is not sufficiently convincing.
2. The explanation of how the qpa-NNs leverage graph structures could benefit from further elaboration to enhance clarity.
3. The scope of the considered noise types is limited, and the authors have benchmarked their results exclusively on small-scale devices.

**Questions:**

1. Have the authors compared their qpa-NN with the so-called “stability baseline model”, (which was mentioned to be better than CNN-based method Hothem et al. [2023c]) ?

**Limitations:**

There does not seem to be negative social impact of this theoretical research.

---

> ### Author Rebuttal · Authors · 2024-08-07
>
> We are grateful for the reviewer's summary of our paper and their recognition of its strengths, especially our superior performance compared to CNNs. Your review and constructive feedback will be instrumental in improving our manuscript, significantly enhancing its clarity and impact.
> In response to your feedback, we plan to include the following revisions in our final manuscript:
>    - A new large-scale demonstration of a qpa-NN trained to predict the process fidelity of 100-qubit circuits run on a simulated quantum computer experiencing low-levels of coherent and stochastic errors. This demonstration conclusively shows that the qpa-NNs scale to large-scale systems even when predicting process fidelity.
>    - An additional paragraph outlining the novelty of our work.
>    - A re-written Section 3.1 that better explains how our qpa-NNs leverage graph structures to reduce their parameter counts and make accurate predictions of a circuit’s fidelity.
>    - Results from three new 4-qubit demonstrations using three new error models.
>
> We now address the four listed weaknesses and explain how our new results/edits address each of the weaknesses. We also include a comparison to the stability baseline model (SBM) from Hothem et. al. [2023c].
>
> **Weakness 1**: the lack of a large-scale demonstration (same as the response to reviewer k1xb).
>
> We agree with the reviewer that the lack of a large-scale demonstration is the main weakness of our paper. We hope that our new 100-qubit demonstration will satisfy the reviewer. We believe that the qpa-NNs’ strong performance on this data conclusively demonstrates the scalability of our approach.
>
> Fig. 1 in the rebuttal material depicts the results from training a qpa-NN to predict the process fidelity of circuits run on a noisy 100-qubit quantum computer experiencing low-levels of weight-1 coherent and stochastic errors. Because of the infeasibility of performing a strong simulation of coherent errors on a 100-qubit device, we used a first-order approximation to the strong simulation, like the technique used in the non-Markovian simulations in [Hothem et. al., 2023]. The qpa-NN achieves a MAE of .097%. This result shows that it is possible to construct qpa-NNs for large system sizes, and that they can achieve excellent prediction accuracy at large system sizes.
>
> **Weakness 2**: novelty.
>
> While we appreciate the reviewer’s concerns about our work’s novelty, we disagree with their assessment. In particular:
>    - Our approach is fundamentally different from past works in that it uses bespoke networks inspired by an in-depth understanding of the underlying physics of quantum computers. This innovation is akin to introducing physics-informed neural networks for solving PDEs or CNNs for image recognition, but for a more specific task.
>    - Moreover, we are the first to apply any kind of neural network to predicting process fidelity. Unlike PST, process fidelity is defined for any quantum gate, circuit, or channel and is the metric of choice for reporting gate and circuit performance. It is also estimated by many popular benchmarking protocols (e.g., randomized benchmarking). Our work thus addresses a widely applicable problem that is of general interest to the quantum computing community.
> We will clarify the novelty of our work by adding a paragraph outlining our contributions in the next draft.
> **Weakness 3**: unclear explanation of how the qpa-NNs leverage graph structures.
>
> Our qpa-NNs leverage a graph structure to greatly reduce their size, enabling the construction of qpa-NNs for modelling many-qubit systems. Our qpa-NNs are scalable because they model a polynomial-sized set of error types, and they use a graph to represent which parts of a quantum circuit a particular error’s rate (at circuit layer
> i) will plausibly depend on (which specifies the input to each that error’s associated subnetwork). This error rate dependency is typically associated with the physical layout of a quantum computer’s qubits, and so we use the graph encoding of this layout to model this dependency. We plan to revise our description of how our qpa-NNs leverage graph structures to better explain this.
>
> **Weakness 4**: limited noise models.
>
> We plan to include results from three new 4-qubit demonstrations to allay the reviewer’s concern that we only evaluated our qpa-NNs on a limited number of error models. Copying from our response to reviewer oCVW:
>
> We performed three new 4-qubit demonstrations to better support our claim that the qpa-NNs improved performance is due to their  ability to model coherent noise. Each new demonstration used the same setup as the original 4-qubit demonstration, but with a different error model. The error models differed in the ratio of total coherent to stochastic noise allowed in each gate’s error model. If we include the original demonstration, we now have results from error models whose ratios range from no coherent noise (maximum H error of 0 and maximum S error of .001) to purely coherent noise.
>
> Our final manuscript will include a broad suite of realistic noise models, ranging from complicated experimental noise models to simple noise models for 100-qubit quantum computers.
>
> **Additional discussion**: comparison to the stability baseline model (SBM).
>
> The qpa-NNs perform favorably compared to the SBM in Hothem et al. [2023c]. Fig. 2 in the rebuttal document shows that the qpa-NN outperforms the SBM on one device, achievers peer performance on three devices, and slightly worse performance on two devices.
>
> However, it is a category mistake to compare the SBM to the qpa-NNs because the SBM is not a predictive model of a device’s capability. The SBM is constructed by rerunning each circuit and comparing the results to those from the original run. Therefore, it quantifies how stable the device is by measuring how a circuit’s fidelity changes over time. Alternatively, it measures how well a device’s past performance predicts its future performance.

---

> > ### Comment · Reviewer_beTU · 2024-08-12
> >
> > Thank you for your detailed response. A good portion of my concerns have been addressed.

---

### Official Review · Reviewer_oCVW · 2024-07-12

**Soundness:** 2
**Presentation:** 3
**Contribution:** 3
**Rating:** 6
**Confidence:** 4

**Summary:**

The paper presents an approach to improve the state of the art in quantum capability learning, which is the task to predict the prowess for error when running a specific quantum algorithm given a fixed quantum computer. The tackle this, the authors introduce some specializations on (graph) neural networks that fit the nature of quantum computers and their errors especially well. The new approach yields better results than purlely CNN-based methods on a specific synthetic and empirical data.

**Strengths:**

The approach is very interesting from both a quantum and a machine learning point of view. The issue of quantum capability learning is described in geat detail and the innovation of the approach becomed clear. In the method to intertwine neural network architectures with their subject of training, especially when that subject is a quantum computer, I see great potential for further investigation.

**Weaknesses:**

To be frank, I consider the issue of quantum capabilty learning (especially as it is motivated in this paper) a rather artificial one, given the quite small number of potentially useful quantum algorithms. And I think this leads to some rather weak arguments, mostly in the motivation of this paper. However, quantum capability learning (as the authors examine it in this paper) is still an interesting and relevant topic. I suggest cutting down on practical promises and focusing on the pure challenge of predicting a quantum computer's behavior using classical neural networks, which I consider a sufficient motivation for the study of this topic.

While the performance-based analysis has good results, I would have wished for a more in-depth take that can actually clarify the speculation posed in the introduction: "Our qpa-NNs' improved performance is likely largely due to their improved ability to model the impact of coherent errors..."

**Questions:**

None.

**Limitations:**

No comments.

---

> ### Author Rebuttal · Authors · 2024-08-07
>
> We are grateful for the reviewer's summary of our paper and their recognition of its strengths. We are also pleased that the reviewer finds modelling a quantum computer's performance with neural networks to be an interesting and worthwhile problem. Your detailed review and constructive feedback will be instrumental in improving our manuscript. Implementing your suggestions will significantly enhance our paper's quality and impact.
> In response to your feedback, we plan to make the following changes to our paper:
>    - Include new 4-qubit demonstrations on simulated quantum computers experiencing different ratios of coherent to stochastic error rates.
> We now address, in order, the two listed weaknesses and explain how our new results/edits address each of the weaknesses.
>
> **Weakness 1**: lack of motivation behind quantum capability learning.
>
> We respectfully disagree with the reviewer’s assertion that the motivation for quantum capability learning is weak given the small number of useful quantum algorithms. We believe that quantum capability learning will be especially important in the early fault-tolerant era as devices grow beyond our abilities to classically simulate, while remaining too noisy or small to reliably implement general quantum algorithms. It is precisely in this early fault-tolerant era when we will need accurate, scalable, and fast-to-query predictive models of a quantum computer’s capability to better understand which experiments to run and which devices to build. Nonetheless, we are happy to read that the reviewer believes that the “pure challenge of predicting a quantum computer’s behavior using classical neural networks” is relevant and sufficiently motivating.
>
> **Weakness 2**: failure to substantiate the claim that the qpa-NNs outperform the CNNs due to their improved ability to model the impact of coherent errors.
>
> We performed three additional 4-qubit demonstrations to better support our claim that the qpa-NNs improved performance is due to their improved ability to model coherent noise. Each new demonstration used the same setup as the 4-qubit demonstration in the original draft except using a different error model. The error models differed in the ratio of total coherent to stochastic noise allowed in each gate’s error model. If we include the original demonstration, we now have results from error models whose ratios range from no coherent noise (maximum H error of 0 and maximum S error of .001) to purely coherent noise.
>
> As shown in Fig. 3, as we increase the ratio of coherent to stochastic errors, the CNNs’ performances begin to diverge from the qpa-NNs' performances, before ultimately stabilizing at a statistically significant worse prediction accuracy. These results confirm our claim that the qpa-NNs outperform the CNNs, in part, because of their improved ability to model the effect of coherent errors on process fidelity.

---

> > ### Comment · Reviewer_oCVW · 2024-08-14
> >
> > I do not agree with the points made here.
> >
> > Most importantly, the fact that the qpa-NN's performance scales better with quantum noise compared to a classical approach still is not a sufficient argument that handling quantum noise is the reason for that performance.
> >
> > I advise to be cautious about making too grand claims in either case.

---

### Author Rebuttal · Authors · 2024-08-07

We sincerely thank the referees for their time and insightful comments on our manuscript. Your thorough review and constructive feedback have been invaluable in identifying areas for improvement and clarity. We are confident that incorporating your suggestions will significantly strengthen the quality and impact of our paper, making it worthy of inclusion at NeurIPS 2024.

In response to each reviewer’s feedback, we plan to modify our paper to:
   - Better support our claim that our qpa-NN approach scales by including a new large-scale demonstration on a simulated 100-qubit computer.
   - Better support our claim that the qpa-NNs’ improved performance is due to their improved modeling of coherent errors by adding an appendix with results from three new simulated 4-qubit devices, each experiencing different ratios of coherent to stochastic error strengths.
   - Clarify how our approach differs from, yet synergizes with, mirror circuit fidelity estimation (MCFE) [Proctor et. al., 2022] by adding a few sentences to Section 2.2.
   - More clearly explain how our approach avoids the exponential scaling afflicting other methods for predicting process fidelity by rewriting Section 3.2.
   - Clarify how our qpa-NNs exploit the graph structure of a quantum computer’s native connectivity graph to efficiently track and model the effects of only the most relevant errors by rewriting Section 3.1.
   - Better explain the novelty of our results by adding a paragraph to the introduction clearly stating this work’s novel contributions to the literature.

Additionally, we have provided a comparison between the qpa-NNs and the stability baseline model (SBM) in [Hothem et. al., 2023c] in our rebuttal. We chose not to include these results in our revised paper as the SBM is not a predictive model and its inclusion would muddle the paper’s presentation. Nonetheless, we strongly believe that our revisions comprehensively address each reviewer’s critiques and that our final product will merit inclusion in this year’s NeurIPS. Lastly, we include an attached pdf with supporting figures.

We now briefly outline how our proposed revisions address specific reviewer critiques. We provide full discussions in our responses to each individual reviewer.
   - Our 100-qubit demonstration should allay reviewer beTU’s and k1xb’s concerns about analyzing only small-scale devices. As shown in Fig. 1 of the rebuttal PDF, our trained qpa-NN obtained a mean absolute error of .097% when predicting the process fidelity of 100-qubit circuits run on a simulated 100-qubit quantum computer experiencing weight-1 coherent and stochastic errors. Further details are provided in our rebuttals to each reviewer. Unfortunately, it is not possible to perform a 100-qubit demonstration on real hardware as IBM’s cloud-accessed, 127-qubit processors are currently too noisy to reliably execute circuits with high fidelity.
   - New 4-qubit simulations should satisfy reviewer oCVW’s desire for a more thorough investigation of whether the qpa-NNs’ improved performance is due to their improved ability to model coherent errors, as well as reviewer beTU’s concern about the scope of the considered noise types. In our new 4-qubit demonstrations, we repeated the 4-qubit demonstration in the paper using three new error models, each with a different ratio of coherent to stochastic errors. As shown in Fig. 3, as we increase the ratio of coherent to stochastic errors, the CNNs’ performances rapidly deteriorate, while the qpa-NNs’ performances ultimately stabilize. These results confirm our claim that the qpa-NNs outperform the CNNs, in part, because of their improved ability to model the effect of coherent errors on process fidelity.
   - Revisions clarifying how our approach differs from, yet synergizes with, MCFE address reviewer k1xb’s request for clarification. Our revisions more clearly explain the different goals of our qpa-NN approach—to build predictive models of a quantum computer’s capability that can predict a circuit’s fidelity without running the circuit—and MCFE—to estimate the fidelity of select circuits by running those circuits on the quantum computer. Our revisions also clarify how to use MCFE to efficiently gather training data for the qpa-NNs, a necessary step in a scalable pipeline for building predictive models.
   - Revisions clarifying our approach to predicting process fidelity address reviewer k1xb’s concern that our approach may have scalability issues. In our revisions, we explain how we avoid the exponentially expensive cost of evaluating equation 3 by focusing solely on predicting the effect of the most important errors in a quantum computer—local, low-weight errors—on process fidelity. Because the number of local, low-weight errors scales polynomially with device size, our qpa-NNs’ parameter counts also scale polynomially with device size. Put another way, we design qpa-NNs to approximate eq. 3 by only predicting a polynomial-sized set of the most important terms in equation 3.
   - Revisions clarifying how our qpa-NNs use a graph structure to encode the spatial dependencies of a quantum computer’s error structure, and how this enables our qpa-NNs to have a small, polynomial number of parameters.
   - Revisions more clearly stating our contributions address reviewer beTU’s concern that our approach is not “sufficiently novel.” As explained in our rebuttal, we introduce a novel, sophisticated neural network architecture for solving the quantum capability learning problem and are the first group to tackle the problem of predicting process fidelity, the most widely used circuit-success metric in the quantum computing community.

---

### Decision · Program_Chairs · 2024-09-25

**Decision:**

Accept (poster)

**Comment:**

The paper introduces a neural network to predict quantum capability, namely quantifying the noise in the quantum circuit. The work focuses on Clifford gates which can be simulated efficiently and presents experiments on quantum devices as well. The method improves the state of the art for predicting the behaviour of a quantum computer using classical neural networks, an important direction in quantum computing. The rebuttal addresses some of the questions on scalability and related work.